# Unlocking Horse Y Chromosome Diversity

**DOI:** 10.3390/genes13122272

**Published:** 2022-12-02

**Authors:** Irene Cardinali, Andrea Giontella, Anna Tommasi, Maurizio Silvestrelli, Hovirag Lancioni

**Affiliations:** 1Department of Chemistry, Biology and Biotechnology, University of Perugia, 06123 Perugia, Italy; 2Department of Veterinary Medicine, University of Perugia, 06126 Perugia, Italy; 3Department of Biology and Biotechnology “L. Spallanzani”, University of Pavia, 27100 Pavia, Italy

**Keywords:** horse Y-chromosome variation, horse genetic diversity, uniparental genetic markers, worldwide horse breeds, horse MSY phylogeny

## Abstract

The present equine genetic variation mirrors the deep influence of intensive breeding programs during the last 200 years. Here, we provide a comprehensive current state of knowledge on the trends and prospects on the variation in the equine male-specific region of the Y chromosome (MSY), which was assembled for the first time in 2018. In comparison with the other 12 mammalian species, horses are now the most represented, with 56 documented MSY genes. However, in contrast to the high variability in mitochondrial DNA observed in many horse breeds from different geographic areas, modern horse populations demonstrate extremely low genetic Y-chromosome diversity. The selective pressures employed by breeders using pedigree data (which are not always error-free) as a predictive tool represent the main cause of this lack of variation in the Y-chromosome. Nevertheless, the detailed phylogenies obtained by recent fine-scaled Y-chromosomal genotyping in many horse breeds worldwide have contributed to addressing the genealogical, forensic, and population questions leading to the reappraisal of the Y-chromosome as a powerful genetic marker to avoid the loss of biodiversity as a result of selective breeding practices, and to better understand the historical development of horse breeds.

## 1. Introduction

The horse (*Equus caballus*) is one of the domestic species to have played an important role in the development of human society. Over the centuries, their domestication necessarily resulted in a strong pressure in the selection of individuals and breeds [1,2,3,4]. The adaptation of horses to the human niche led to the current genetic variation [5,6] carried by the domestic horses that left the Western Eurasian steppes at the beginning of the second millennium BC and moved towards Eurasia, thus almost entirely replacing the wild populations [7].

Methodological and bioinformatics tools have recently been developed, allowing for increased accuracy in the analysis of high-throughput genomes, and over last decades, the equine research community has aimed to reconstruct the evolutionary paths that can still be detected in their genomes [8]. Genetic evidence has pinpointed multiple horse domestication events occurring across Eurasia 5000–6000 years ago [2,9,10,11,12,13,14,15,16]. After this evidence emerged, the history of the horse domestication process was revised. Furthermore, the analysis of the horse remains from the Botai site, deeply described by Outram and colleagues [13], identified this area as the cradle of extant Przewalski’s horses’ ancestors [17], while modern horses have been domesticated in a more Western centre: the lower Volga-Don region [7]. The first appearance of the ancestor of all modern horses is dated back to 4200 years ago [18]. Since then, both the disappearance of the earliest domestic lineages and the emergence of the modern ones occurred, with an increasing genetic variability that remained constant during the last 4000 years until it significantly dropped in the last ≈250 years [19]. These modern horse lineages rapidly spread across Eurasia, colonizing a region from central Anatolia to central Russia, completely replacing almost all other local populations about 4000 years ago [18]. The genetic profile of these colonizing horses was found in the archaeological remains buried in Sintashta kurgans in the West-Eurasian steppe [20,21].

The improvement of livestock breeds usually involves inbreeding to select individuals with favorable traits [22]. Recently, many studies have been focused on the loss of biodiversity [23,24,25] or the increase in deleterious genotypes [26] caused by such inbreeding (as reviewed in [7]). In the last 200 years, the inbreeding practice led to the phenotypical expression at homozygous sites of deleterious variants [27], as highlighted when the coefficients are calculated in order to evaluate the genetic diversity among different horse breeds [28,29,30,31,32,33,34,35,36,37,38,39,40,41,42,43,44,45,46,47,48,49,50,51,52,53,54,55,56,57,58,59,60,61,62,63,64,65,66,67,68,69,70,71,72,73,74,75,76,77,78]. Inbreeding has caused reduced fertility and survival among offspring of related individuals, resulting in a decline in fitness [79] and the emergence of disadvantageous traits [27,80]. The increased knowledge about inbreeding depression and the genetic structure of breeds [81,82] have allowed breeders to select horses by avoiding mating closely related individuals [83].

The risk of losing genetic diversity, resulting in more uniform populations with highly specialized traits, is especially evident in those breeds that are under strong human selection [26]; thus, it is recommended that breeders use the less intensive practice of line breeding and ensure a certain extent of variation among horse breeds [84]. As a result of intensive breeding programs, domestic horse populations changed along with the human development, above all in the last 200 years [19], as highlighted by most research focused on autosomal loci or maternally inherited mitochondrial DNA (mtDNA) (as reviewed in [85]).

## 2. The Horse Genome

In 1995, the international consortium of the Horse Genome Project was established to enhance knowledge about the evolutionary history and inherited traits of domestic horses, supported by many funding organizations. The project’s team of researchers decoded the diploid genome of a domestic horse named Twilight, an English Thoroughbred female racehorse, and found that the genome is distributed over 31 pairs of autosomes, the X chromosomes and the mitochondrial genome.

The RefSeq genome records for *E. caballus* were annotated by an automated pipeline (NCBI Eukaryotic Genome Annotation Pipeline) on the only two high-quality genome assemblies for equids: EquCab2.0 (accession number: GCF_000002305.2) [86] and EquCab3.0 (accession number: GCF_002863925.1) [87].

The first reference sequence (EquCab2.0) was obtained by performing a Sanger sequencing with a 6.8-fold genomic coverage and including about 315,000 BAC clones from a library collected from Twilight’s half-brother, Bravo [88], and partial sequences from seven horses belonging to different breeds. Over a million of SNPs were identified and used to perform molecular, evolutionary and clinical studies on horses [89].

As EquCab2.0 contained many gaps, the genome of Twilight was re-sequenced and assembled in 2018 using high-throughput sequencing technologies, thus resulting in the new reference genome: EquCab3.0. The new assembly contains 3771 gaps comprising 9 Mb (0.34% of the genome) with a scaffold N50 of 86 Mb (Table 1) [87,90].

### Horse Y Chromosome Sequencing and Comparison with Humans and Other Mammals

However, the resulting horse reference genome was still incomplete as it was based on the genome of a female horse; that is, the analysis of the horse Y chromosome was lacking. During last ten years, many efforts have been made to produce Y chromosomal DNA data for *E. caballus* [16,91,92]. Until 2018, Janečka and colleagues published the assembly of 9.5 Mb based on the sequencing of the Y chromosome from the thoroughbred stallion Bravo, thus providing the first comprehensive assembly of the male-specific region of the Y chromosome (MSY) (accession number: MH341179) [93].

Several special features set the Y chromosome apart from the rest of genome: its male-limited transmission, the absence of recombination, abundance of Y-specific repetitive sequences, degeneration of Y-linked genes during evolution, acquisition of autosomal genes, and accumulation and functional cluster of “testis genes” for maleness and reproduction. The recent advent of new molecular tools in genomics shed light on the biological and medical relevance of the Y chromosome and helped answer specific biological queries about the roles of the Y chromosome in testis determination, spermatogenesis and beyond the reproductive tract, with a large implications on health and disease [94].

The human Y chromosome was sequenced in 2003 [95], followed by the Y chromosomes of chimpanzee, mouse and rhesus macaque [96,97,98]. To date, additional Y chromosome genes have been mapped and/or functionally analysed in many other species, from insects [99] to carnivores [100] and cattle [101,102], and to a limited extent for other domesticated species [103,104], often targeting specific questions.

Comprehensive comparative genomic analyses of the Y chromosomes of multiple mammalian species have demonstrated that, despite their shared ancestry in terms of evolutionary history, mammalian Y chromosomes display enormous variation in size, gene content, and structural complexity among species. Several unique features of the Y chromosome, such as the opportunities for crossing over restricted to the pseudoautosomal regions, the functional specialization for spermatogenesis and the high degree of sequence amplification of repetitive DNA, have contributed to this wide variation [94]. Nevertheless, the species studied until now showed a progressive genetic decay in the MSY’s diversity resulting in deletions and gene losses that collectively decimated the Y chromosome [105]. In horses, even though the stallion fertility has prime importance in breeding management, very little is known about the complexity of the equine Y chromosome’s structure and its genetic degeneration.

Genomic analyses recently showed several autosomal loci and mutations significantly associated with stallion fertility [106], and abnormalities in a number of X- and/or Y chromosomes have been shown to be causes for aberrant sexual development [107]. As previously reviewed [83], among all the above mutations associated with infertility, only the deletion in the sex-determining region (SRY) leads to infertility in horses, which was found in the 25% of horses with chromosomal aberrations, but it was not detected in other species [108]. All other alterations and rearrangements known to be associated with horse infertility are ascribed to loci not localized in the Y chromosome. Currently, the growing deep sequencing and SNP genotyping array is likely to increase the number and complexity of chromosomal syndromes associated with infertility detected in horses, especially those related to Y-chromosomal abnormalities, where the complex genetic architecture is still understudied [82,109].

The exclusion of the Y chromosome from genomic analyses may previously have been justifiable, based on the assumption that it was a genetic wasteland, a junkyard, a barren chromosome [110]. Nowadays, the increase in published studies on the topic makes clear the potential of analyses focused on the variation in the Y chromosome for identifying individuals with greater susceptibility to disease and for forensic analysis and paternity testing. As argued by Chowdhary and other researchers, “Finally, the Y chromosome got some stardom and was not just referred to as the sex-determining chromosome” [111,112].

Nevertheless, the horse Y chromosome remains the most understudied chromosome in the equine genome. It is comparable in size to the smallest equine autosomes, around 40–50 million base pairs (Mbp). A very limited effort has been made to date to develop a gene map for this chromosome and understand its structure and organization [113].

In 2018, Janečka and colleagues generated the first comprehensive assembly and functional annotation of the euchromatic male-specific region of the Y chromosome (eMSY). They demonstrated its dynamic nature and provided a reference sequence useful in improving our knowledge about the male equine development and fertility [93]. In a radial plot denoting the presence or absence of a gene on the Y chromosome, Janečka and collaborators recorded 88 unique genes and transcripts and compared them across 13 mammalian species (12 from Eutheria and one from Metatheria). The horse was the most represented with 56 documented genes (56% of MSY), among which there are five pseudoautosomal genes and no pseudogenes, followed by human and pig, with published evidence for 38% of the MSY in these species (Figure 1 and Appendix A).

The conservation of equine MSY was at higher level [93] than that previously observed in primates [96,97]; indeed, even though the divergence time between horse and donkey [114] is comparable to that between human and chimpanzee [96], they show significantly different evolutionary rates in the MSY. Horse and donkey retain the same single-copy MSY and multi-copy sequences [93,113], while more than 30% of human and chimpanzee MSY genes are not homologous and differ in gene content [96].

## 3. The Debate about Stallion Pedigrees

In contrast to the high mtDNA variability reported in previous studies [2,8,9,115,116], which was already present immediately after their domestication [17,19,117], the Y chromosome shows a very low level of genetic polymorphism in modern horse populations [91,118]. Despite a large diversity of domestic male founders contributing to their early domestication [117], the Y chromosome variability considerably decreased in the last 200 years because of selective pressures and the reduction in the stallion population size operated by breeders [19]. It is known that Thoroughbreds in particular were widely used in the development of many breeds to obtain a less bulky and lighter horse more suitable for riding [119].

Due to the intensive selection of stallions, the establishment of a studbook-mediated sire line represents the main cause of this lack of Y-chromosome variation. The importation of stallions used to improve autochthonous breeds started from the Arabian region, where the modern legacy is represented only by few foundation sires, thus confirming the strong sex bias in horse breeding [26]. In any case, there are significant differences among the quality of breeding information depending on the breed.

It is common knowledge that pedigree has been used in breeding programs as a predictive tool for the unique and reliable identification of individual organisms in breeding management and conservation since to the late 18th century [120]. Pedigree data were already employed to analyse the population’s structure and identify factors that affected the genetic variability of horses [121].

The pedigrees of Arabian Horses, English Thoroughbreds and some European breeds [61,70,75,77,84,111,122] have been deeply described in the dedicated studbooks, showing a notable influence of Arabian, Spanish and Thoroughbred stallions. However, the information included in studbooks is not always error free, and genetic analyses are essential for an accurate characterization of patrilines in a breed [77]. Indeed, for horses, as for many other species, it is crucial to know the animal’s founders; thus, today, the accuracy of pedigrees recorded in modern horse studbook/registries is based on the parentage verification [123].

Through the molecular analysis of DNA markers, it is possible to depict the genetic variation of a breed and allow for the identification of errors in the pedigree registration [124,125]. These analyses could ensure the accuracy of studbook data and prevent the loss of genetic diversity, thus representing a crucial factor for the preservation and management of indigenous breeds. In any case, the results of molecular analysis depend on the sample received from individuals which could tamper with the material to have a specific result [123]. As many horse breeds were improved with the Arabian Horse, this breed was subject to accurate molecular analyses. The examination of nearly 400 Arabians and related breeds revealed extensive outcrossing to the English Thoroughbred, even though the pedigrees reported an entirely Arabian ancestry [126]. This is only one example of the frauds that move around the horse pedigrees with economic and race-related motivations. Nevertheless, the genomes of these horses unveiled a particular genetic signature probably representing the common ancestor of both Arabian and English Thoroughbred horses. However, no sampled modern population has provided answers for the places and times of origin of the ancestral population [126]. The Y-chromosome sequence data date back the origin of modern Arabian horses above all to crosses of mares native to the British Isles with three oriental stallions (Godolphin Arabian, Darley Arabian and Byerly Turk), although Thoroughbreds seems to share ancestry with Akhal-teke horse, the remnant of the Turkoman horse [91]. Furthermore, the contribution from these three stallions to the genome of an English Thoroughbred is probably overestimated [127], and even if Arabian and English Thoroughbred breeds share a common ancestry that has not been deepened, there is a small proportion of Arabian DNA in the modern Thoroughbred [126,128]. The identification of the prohibited outcrossing (a practice of crossing between different breeds made to introduce distantly related genetic material into a breeding line) in most of Arabian samples created many controversies among breeders and buyers [123]. Thus, they cannot count on the pedigree of all modern horses to be accurate, and the matter does not concern the qualities of the animal, but rather the violation of breed definitions and the tampering of registries. Genetic tools could rapidly improve the accuracy of studbooks and registries, tell the history of each horse and also acquire a cultural significance only if based on the integrity of all subjects involved in the equine industry [123].

## 4. The Horse Y Chromosome’s Variation

Until recent times, the main problem with the horse paternal marker was the lack of sequence polymorphisms [129], and only a few patrilines were suggested in horse domestication [1,130,131].

When the first microsatellite sequences for the equine MSY were detected [130,132] and used to analyse the variability among horse breeds from different continents, no variants were individuated in modern breeds, except for two alleles detected in one polymorphic site (YA16) of native Chinese horses [133]. Later, the sequencing of 4 kb of Y chromosome DNA derived from nine ancient remains (one 2800-year-old domesticated horse and eight wild horses) retrieved in permafrost sites of Siberia and North America, and the comparison with the known Y-chromosomal polymorphisms in modern domestic and Przewalski’s horses, allowed for the identification of 28 segregating sites and eight different haplotypes, thus confirming a genetic variability among horses before domestication also in the paternal line [134] and stimulating new research in this field. In 2017, Librado and colleagues analysed the genomes of 14 horses lived from Bronze to Iron Ages, confirming the presence of a large genetic diversity in the early stages of domestication [117]. Recently, the presence of the most ancient paternal lines in Chinese native horses was confirmed through the analysis of 16 Y-chromosomal microsatellite markers, thus reporting for the first time seven novel microsatellite loci [135], in addition to those previously detected [130,132,136]. Among them, four were highly polymorphic only in Chinese local populations and the analysis of the genetic relationships between all the 268 male horses analysed allowed for the identification of 19 Y-chromosomal haplotypes. Three indigenous breeds (Debao pony, Guizhou and Jinjiang) living in the isolated regions of southwestern China, showed haplotypes distant from those arbored by other Chinese indigenous and introduced horses, thus suggesting a highly paternal diversity preserved in this geographic area [135]. In the last twenty years, many modern horse breeds from different geographic areas were analysed for the Y chromosome by employing different sequencing technologies (Appendix A).

In some cases, the MSY from modern horse breeds was compared with the results from Przewalski’s horse [52,91,111,133,136], which is considered the feral descendant of the domesticated Botai horse [17] and has been thoroughly investigated; as a result, a certain number of variable sites were individuated, and a horse Y-chromosomal phylogeny was generated [84,91,111].

In 2013, Wallner and colleagues sequenced Y-chromosomal BAC clones to obtain a systematic screen for horse Y-chromosomal variants, and described, for the first time, the relationships among the identified haplotypes [92]. They observed a strong influence from the Near East, with the description of six MSY haplotypes for modern horses and two additional haplotypes from the Przewalski’s horse [137]. Among these, three haplotypes (HTs) were particularly common: HT1 represents the ancestral haplotype and was distributed across almost all breeds analysed; HT2 was frequent among all breeds, except for those from Northern Europe and the Iberian Peninsula; HT3 was present above all in the English Thoroughbred [92]. All the remaining haplotypes arose from HT1 and were found only in Northern European breeds. The widespread presence of HT1 and HT2 is probably due to the Oriental wave, as Arabian horses carrying these MSY haplotypes were imported into Central European studs, while the frequency of HT3 in modern horses is attributable to the intensive use of English Thoroughbred in the improvement of other horse breeds. The SNPs and indels identifying the different haplotypes were further investigated in 42 breeds from different areas of Europe, Asia and America [136] and in 13 Chinese indigenous [138], 3 Sardinian [139] and 1 Kazach [140] horse breeds (Figure 2 and Appendix A).

HT1 is the most represented haplotype in the Eurasian context, followed by HT2, which is also particularly frequent among American horse breeds. In particular, differently from the European and North American breeds [92,136], almost half of Chinese horses did not show a direct link between the SNP T/A and the deletion in np 10,594 for the HT3 [138]. The same low variability in the other modern European breeds was observed in three local breeds from Sardinia (Italy), each showing one distinctive haplotype (HT1, HT2 and HT3) frequency: 50% of HT1 in the Sarcidano breed, 75% of HT2 in the Giara breed and 58% of HT3 in the Sardinian Anglo-Arab breed [139]. These haplotype distributions highlight the replacement of autochthonous Y chromosomes with the import of stallions belonging to three paternal lines (ancestral line, Neapolitan/Oriental and Thoroughbred wave) for the improvement of native breeds. Contrary to these findings, Chinese indigenous horse populations seem to have preserved their genetic diversity because they have not been subjected to intensive selection, thus presenting a particular genetic pattern and unique MSY haplotype variants [133,135,138,141,142]. A recent study focused on a large sample of Chinese horses filled the known gap in the worldwide context by adding information on the Asian populations not previously analysed [142]. The screening of single-nucleotide polymorphisms (SNPs), copy number variants (CNVs) and allele-specific CNVs showed a deep genetic distance between Chinese indigenous horse populations and those from Europe and the Middle East.

Two microsatellites investigated in Hucul and Mongolian horses (YP9) and in a Shetland pony (YN04) [136], in addition to the analyses of Y-chromosome-specific high-resolution haplotyping, have revealed even more variation among European domestic horses [91] and the SNP at locus YA16 in the Yakutian Horse [143,144], thus confirming the Y variability within this breed and the preservation of the autochthonous variation [143].

In 2017, Wallner and colleagues partially assembled Y-linked regions and generated a horse MSY reference sequence of 2491 high-quality single-copy contigs, covering a length of 1.46 Mb [91] of the 15 Mb-spanning euchromatic part of the horse MSY [113,145]. Then, they mapped the whole-genome data of 52 male domestic horses from 21 breeds, a Przewalski’s horse and a donkey, to detect their MSY variants and build a phylogenetic tree using the Przewalski’s horse and donkey as outgroups [91]. They observed that almost all MSY haplotypes of modern equine breeds clustered together in a predominant group, called “crown group” (A, L, S and T haplogroups—HGs) whose most recent common ancestor originated about 1000–2000 years ago from Oriental founder stallions [143]. This group separated from Northern European horses around 1300 years ago (haplogroup N, found in Norwegian Fjord Horse, Swedish Coldblood horse and Shetland Pony) and from the sister clade I (found in the Icelandic horses) around 1000 years ago [91]. However, all modern European breeds clustered together in the crown group and exhibited an influence of Oriental stallions with the MSY ancestry belonging to the Original Arabian and Turkoman lineages, with the latter profoundly influencing English Thoroughbred stallions (Darley Arabian, Byerley Turk and Godolphin Arabian) to which the sub-branches of haplogroup T have been attributed [91,144]. Then, another 211 variants and 58 haplotypes were described by the analysis of 5.8 Mb of MSY in 130 domestic horses and nine Przewalski’s horses, confirming the clear separation between the crown group (found in Central and South European, North American and most East Asian modern horses) and the non-crown haplotypes (found in Przewalski’s horses and some North European and Asian breeds) [111].

The major MSY clades that belong to the crown group are A (first described in an Arabian Horse), H (first described in a Spanish Horse), L (first described in Lipizzan) and T (first described in Thoroughbreds), while the non-crown group is mostly represented by I (found in Icelandic Horse), J (in Jeju Horse), N (typical of North European breeds), O (found in Mongolian Horse), P (in Przewalski’s horse), and Y (found in Yakutian Horse) clades [84,91,111]. In order to define if a sample belong to the crown group, two key variants (rAY and rAX) have to be tested; then, through the analysis of rA, rW and fYR variants, it is possible to cluster horses into one of the three major crown clades (T, A or H, respectively) [84].

To summarize, among the 169 worldwide horse breeds analysed until now for the Y chromosome (Appendix A), 46 were classified into haplogroups [84,91,111,143] (Appendix A). As reported in Figure 3, 38 breeds belong to the crown group, seven to the non-crown group and only one Asiatic breed (Yakutian Horse) represents both (Figure 3 and Appendix A).

In a worldwide context, the non-crown group was detected in Icelandic Horse, Jeju Pony, Mongolian Horse, North Swedish Draft, Norwegian Fjord Horse, Przewalski’s Horse and Shetland Pony (Figure 3), which, similar to many other native breeds, carry specific breed variants that were not replaced by the Arabian and Turkoman lineages [84].

Due to the large use of Arabian stallions in the improvement of local horse breeds, the recent study by Remer and colleagues also focused on the breeding history of these horses [84]. In the last two centuries, the so-called “Arabian wave” has profoundly influenced the selective breeding practices of European horses [19], but no genetic signatures were found in the English Thoroughbred’s MSY ancestry [126]. A particular genetic variability was found among the Arabian stallions from the Middle East [29,84,126,146], thus suggesting an origin from this area for this breed.

Recent studies focusing on ancient samples tried to describe the times and modes of the loss of horse MSY variation, and different scenarios were proposed. The correlation between the genetic admixtures observed in European populations and the spread of the Yamnaya culture from the Pontic-Caspian steppe [21] initially suggested this area as the centre of horse domestication [91]. The loss of horse Y chromosome diversity originated from there, with one MSY lineage that gradually replaced all the other [118], except for the lineage found only among Yakutian horses [143,144,147] and a quite marked genetic diversity observed today in other Asian breeds [38,140,144,148]. In 2018, Wutke and colleagues conducted an analysis on the MSY polymorphic sites of 96 European ancient stallions dated back from Copper and Bronze Age to the Middle Ages, declaring that the loss of different Y chromosome lineages in modern horses is due to an artificial selection started in the Iron Age and not to a founder or demographic effect [118]. The reduction in horse Y chromosome diversity over time was stated in a study focused on about 1500 MSY polymorphic sites of 105 ancient stallions dated back from the Upper Palaeolithic to Early Modern periods, showing that the genetic diversity of paternal lines decreased during the last 2000 years [19]. In 2021, Librado and colleagues replaced this scenario showing a high genetic diversity before the spread across Eurasia, with horses migrating from the lower Volga-Don region [21] and preserving a quite constant Y chromosome diversity during the last 4000 years, until an important decrease started ≈250 years ago with the intensive breeding programs operated by breeders [19].

## 5. Conclusions and Perspectives

The highly repetitive structure of Y chromosome makes its sequencing and assembly very difficult [149], but the longstanding debates about origin, spread and genetic variability of domestic horses and the abundant studies focused on the female counterpart have allowed researchers to uncover the evolutionary processes that affected the paternal lineages. To enhance knowledge about the evolutionary history and inherited traits of domestic horses, the analysis of high-throughput genomes led to the production of two high-quality genome assemblies for equids (EquCab2.0 and EquCab3.0) [86,87] and the first comprehensive assembly of the MSY [93], representing one of the most complete MSYs for eutherian mammals and filling a gap in the horse genome reference sequence. These results provided an important model for the research focused on stallion biology. Therefore, the recent development in the fine-scaled analysis of the horse Y chromosome has contributed to tracing patrilines and pedigrees [38,91,111,141,143,148] and has provided different scenarios for the time and causes of the loss of Y chromosome diversity [7,19,118]. The discovery of new variants and a better understanding of the pathways through with domestication occurred were possible through the analysis of modern [91,92,93,111,136,138,144] and ancient [7,19,21,118,150,151] horse populations. Despite further investigation still being needed, MSY variation represents a powerful lineage and pedigree tracer crucial for strengthening horse management and is an evaluable genetic marker that contributes to avoiding further loss of biodiversity and understanding the historic development of breeds.

## Figures and Tables

**Figure 1 genes-13-02272-f001:**
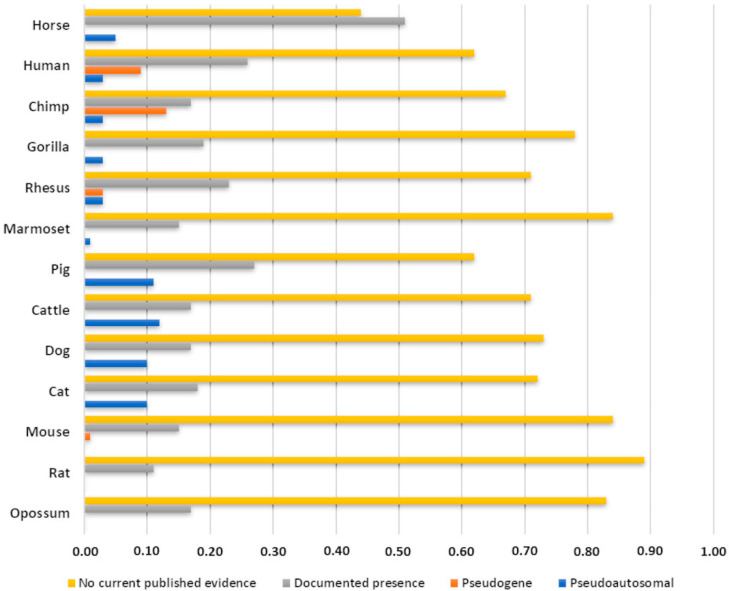
Rate of MSY gene data for 13 mammalian species available in 2018 when the horse Y chromosome assembly was released. Details and comparison were described in [93]. Each bar represents the amount of MSY coding genes, pseudogenes and transcripts known (or not) in horses and other eutherian species with sequenced or partially sequenced MSYs. See also Appendix A.

**Figure 2 genes-13-02272-f002:**
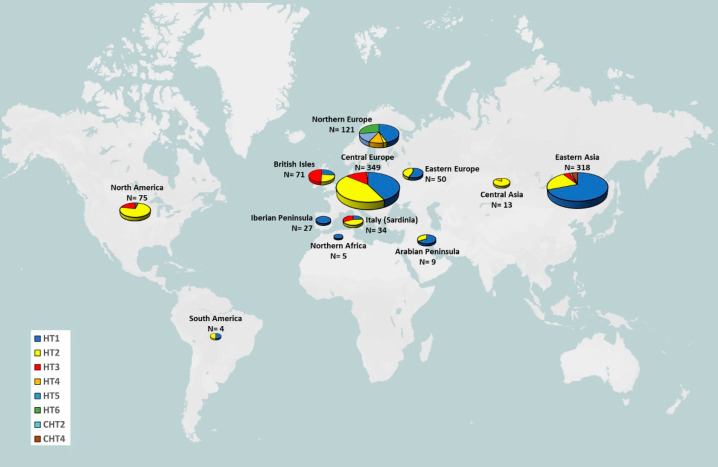
Geographic distribution of Y-chromosome haplotypes (classification as in [92,138]) among 76 modern horse breeds from different geographic areas. Further details are reported in Appendix A.

**Figure 3 genes-13-02272-f003:**
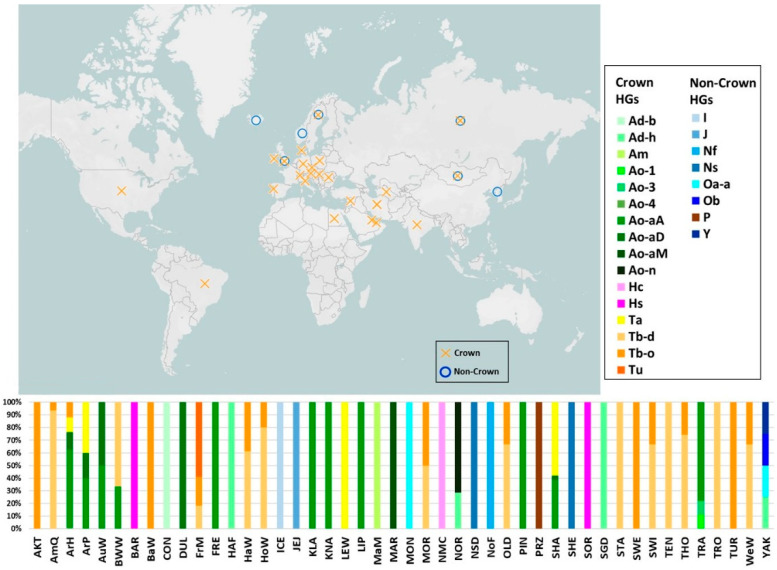
Worldwide distribution of crown and non-crown haplogroups and frequencies of each Y chromosome HG among the 49 breeds analysed in [84,91,111,143]. Geographic origin, breed codes, frequencies and proportion of “crown” and ”non-crown” groups for each breed are reported in Appendix A.

**Table 1 genes-13-02272-t001:** Genome information for reference and representative genomes of horse (EquCab3.0; GCF_002863925.1).

Type	Name	RefSeq	INSDC	Size (Mb)	GC%	Protein	rRNA	tRNA	Other RNA	Gene	Pseudogene
Chr	1	NC_009144.3	CM009148.1	188.26	41.7	4764	4	30	1219	2591	193
Chr	2	NC_009145.3	CM009149.1	121.35	42.2	329	-	-	790	1559	92
Chr	3	NC_009146.3	CM009150.1	121.35	41.0	2541	-	9	666	1333	110
Chr	4	NC_009147.3	CM009151.1	109.46	39.8	2013	-	17	566	1133	100
Chr	5	NC_009148.3	CM009152.1	96.76	40.9	2951	-	69	779	1566	97
Chr	6	NC_009149.3	CM009153.1	87.23	41.5	2868	-	2	603	1431	113
Chr	7	NC_009150.3	CM009154.1	100.79	42.7	3422	-	5	698	1985	222
Chr	8	NC_009151.3	CM009155.1	97.56	42.7	2246	-	6	734	1168	90
Chr	9	NC_009152.3	CM009156.1	85.79	39.9	1378	-	10	491	753	53
Chr	10	NC_009153.3	CM009157.1	85.16	41.7	3009	-	4	660	1637	167
Chr	11	NC_009154.3	CM009158.1	61.68	45.6	3163	-	35	687	1568	62
Chr	12	NC_009155.3	CM009159.1	36.99	45.0	1767	-	9	350	1212	231
Chr	13	NC_009156.3	CM009160.1	43.78	47.0	2034	-	26	426	965	35
Chr	14	NC_009157.3	CM009161.1	94.6	40.6	2016	-	8	538	1041	73
Chr	15	NC_009158.3	CM009162.1	92.85	41.3	1917	-	6	598	1011	58
Chr	16	NC_009159.3	CM009163.1	88.96	42.0	2309	-	5	548	996	52
Chr	17	NC_009160.3	CM009164.1	80.72	38.6	1019	-	5	321	569	62
Chr	18	NC_009161.3	CM009165.1	82.64	39.2	1489	-	4	488	685	53
Chr	19	NC_009162.3	CM009166.1	62.68	39.6	1272	-	2	328	627	62
Chr	20	NC_009163.3	CM009167.1	65.34	41.4	1866	-	196	489	1271	106
Chr	21	NC_009164.3	CM009168.1	58.98	40.3	101	-	2	264	584	61
Chr	22	NC_009165.3	CM009169.1	50.93	45.2	1501	-	1	391	779	26
Chr	23	NC_009166.3	CM009170.1	55.56	40.4	850	-	1	476	544	52
Chr	24	NC_009167.3	CM009171.1	48.29	43.2	1257	-	2	509	740	48
Chr	25	NC_009168.3	CM009172.1	40.28	45.2	1656	-	2	336	775	60
Chr	26	NC_009169.3	CM009173.1	43.15	39.8	698	-	-	197	403	34
Chr	27	NC_009170.3	CM009174.1	40.25	39.5	763	-	1	196	356	23
Chr	28	NC_009171.3	CM009175.1	47.35	42.7	1174	-	3	378	587	26
Chr	29	NC_009172.3	CM009176.1	34.78	40.4	676	-	1	213	342	50
Chr	30	NC_009173.3	CM009177.1	31.4	40.3	527	-	1	195	305	30
Chr	31	NC_009174.3	CM009178.1	26	41.2	446	-	1	130	242	25
Chr	X	NC_009175.3	CM009179.1	128.21	39.3	2087	-	3	514	1206	134
-	MT	NC_001640.1	-	0.02	42.0	13	2	22	-	37	-
Un	-	-	-	97.81	44.5	908	8	8	618	1169	247

## Data Availability

Not applicable.

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
