# Peer review of "Unlocking Horse Y Chromosome Diversity"

_genes, 2022, doi:10.3390/genes13122272_

Round 1

Reviewer 1 Report

Very thorough and demanding overview of the topic!
The only change I would suggest is the 2018 graph (Figure 1.) in the article. Because there have been significant changes in the four years since then!

And a few "ici-pici", I mean smaller than minor, suggestions written into the text!

And you need to decide crown or Crown! Because the original article use it with lower case!

"The crown was postulated to be the footprint of Oriental horses [ 20 ], but the exact ways and timeframes by which the Oriental HTs were disseminated in the past are not yet fully resolved. ‘Non-crown’ HTs have so far been detected only in Asian horses [21,23,24] and in some northern European breeds [20,25]."

Author Response

Very thorough and demanding overview of the topic!

The only change I would suggest is the 2018 graph (Figure 1.) in the article. Because there have been significant changes in the four years since then!

And a few "ici-pici", I mean smaller than minor, suggestions written into the text!

And you need to decide crown or Crown! Because the original article use it with lower case!

"The crown was postulated to be the footprint of Oriental horses [ 20 ], but the exact ways and timeframes by which the Oriental HTs were disseminated in the past are not yet fully resolved. ‘Non-crown’ HTs have so far been detected only in Asian horses [21,23,24] and in some northern European breeds [20,25]."

Authors’ response: we would like to thank the reviewer for the positive comments.

  • l 105. in 2018.
    • Authors’ response: we have added “, in 2018.”
  • l 163. Congratulation! that is really necessary in this type of reviews! ok, "wasteland" is a citation, but "junkyard" is very refresser!
    • Authors’ response: we sincerely thank the reviewer for this comment!!
  • l 201. The [97] was published in 2018. an update is not possibile for this review? because the current dataset could be very interesting! Or sign the publication date!
    • Authors’ response:we have included the publication rate and detailed accordingly the figure legend
  • l 270. Yes, Cunningham's article is very important in this General Stud Book topic! He had acces to the original database!
    • Authors’ response: we would like to thank the reviewer for this specification.
  • Table 2. Lippiza is in Slovenia in Central Europe.
    • Authors’ response: we have deleted this line from the table, as it was an oversight.
  • l 377. Crown? it is seems to be a name in this form with capital! But we use it as name...
    • Authors’ response: we have replaced "Crown" with "crown" throughout the text.

Reviewer 2 Report

Cardinalli and colleagues overview an increasingly interesting topic, addressing the paternal evolutionary history of horses, probably the most important animal for humans.

I think the authors deserve the opportunity to fix my very major concerns. There are some statements that are not true, and some very important papers still need to be commented and referenced:

1. "The adaptation to the human niche and the repeated interactions between domestic populations and surrounding wild relatives led to a continuous gene flow resulted in the current genetic variation [5,6]. "

In our last paper (Librado et al; Nature 2021), we demonstrated that DOM2 horses (the lineage leading to present-day domesticates) carried a diversity of mt and Y haplogroups before their expansion across Eurasia 4200 years ago. They did not interact much with wild relatives after this period. They simply replaced them. Therefore, there was no such continuous gene flow after horses left their homeland.

2. "Genetic evidence pinpointed that multiple horse domestication events occurred across Eurasia 5,000 - 6,000 years ago [2,8–15]".

There was one dead-end domestication event at Botai 5,500 years ago (the so-called DOM1 horses, Outram 2009 Science; Gaunitz 2018 Science), and then another one in the lower Volga-Don region. The horse that originated from the latter expanded across Eurasia 4,200 years ago (Librado 2021; Nature). There were no multiple horse domestication events 5,000-6,000 ya. The horse lineages that were domesticated already had a diversity of haplogroups, which was originally misinterpreted as multiple domestication events. But we know now that this was not true.

3. "with an increasing genetic variability that remained constant during the last 3,000 years". 

We showed that genetic diversity in the autosomes remained more or less constant in the last 4,000 years , until 250 years ago(Fages et al 2019; Cell). If we look at the Y chromosomal diversity, we see it expanded 4,200 years ago as DOM2 horses spread through Eurasia and increased their population size (Librado et al 2021). I do not understand where their sentence, and this 3000 years ago, comes from. 

4. "until it significantly dropped in the last ≈250 years". This statement requires to be backed up by an original reference (eg. Fages et al). Only citing other reviews is not fair to the original publications.

 5. "completely replacing almost all other local populations 3,000 years ago". Again, it should be 4,000 years ago. 

6. "the genetic diversity among horses is very great". What does it mean to have a great diversity? I guess the authors mean "large"? Please clarify

7. "Recently, many studies have been focused on the loss of biodiversity [21–23] or the increase of deleterious genotypes [24] caused by the inbreeding, whose coefficient was calculated to evaluate the genetic diversity among different horse breeds [25–75]."

I think there is one fundamental paper missing here: Orlando and Librado 2019 published in Genes (it analyses the relation between inbreeding and deleterious mutations, as reviewed in Librado and Orlando 2021 (Annual Reviews)).

8. "It was finally released in 2018, with results distributed over 31 autosomes, the X chromosome and the mitochondrial genome"

The horse genome was first released in 2009 (Wade et al.; EquCab2). In 2018, we had an improved version (EquCab3). Please clarify your text. It is confusing. It seems that the first horse assembly was produced in 2018.

9. Their phrasing of this whole section is simply too similar to Kalbfleisch et. al 2018 (the paper presenting EquCab3), which is not acceptable.

10. "As EquCab2.0 contained many gaps, the genome of Twilight was recently re-sequenced, through Next Generation Sequencing (NGS) techniques".

EquCab3 not only required NGS techniques, but also newer sequencing technologies such as PacBio, proximity ligation libraries, etc. 

11. "Contextually with the release of EquCab3.0, Janečka and colleagues published the assembly of 9.5 Mb based on the sequencing of the Y chromosome from the Thoroughbred stallion “Bravo”, a half‐brother of Twilight, thus completing the Thoroughbred‐based horse reference genome and providing the first comprehensive assembly of the male‐specific region of Y chromosome (MSY) (accession number: MH341179)"

I think Lippold et al (2011; Nature Communications), Wallner et al. (2013; PLoS One), Wallner et al (2017; Current Biology) deserve to be cited in this context as well. They are cited later in the context of Y-chr diversity, but they also generated the earliest scaffolds of the Y chromosome. 

12. I would considerably reduce the section comparing the Y chromosome of horse to that of other mammals. Many paragraphs are simply descriptive, and provide nothing new conceptually. This is true in other sections, but it makes the reading particularly boring here. We have databases for consulting how many genes there are, and other features of the Y chromosome. Less is more sometimes.

13. "In mammals, the Y chromosome is characterized by the male-specific and nonrecombining portion (MSY), inherited only along paternal lines from a father to his colts, thus becoming a large-used marker in equine studies focused on the origin of stallions [136]." 

I do not think we need this sentence again at this stage of the review (last section). We got what the Y chromosome is already.

14. "Later, Lippold and colleagues sequenced 4 kb 291 of Y chromosome of DNA derived from nine ancient (one 2,800-year-old domesticated and eight wild horses) remains retrieved in permafrost sites of Siberia and North America [142]."

You need to talk about Librado et al. 2017 (Science) here.

I stopped reviewing in detail their work here. There are simply too many errors to keep going as a reviewer. I am keen on recommending this for publication, but please be correct, and unambiguous your own phrasing. Reduce only-descriptive sections (eg. how big is the assembly, how many genes, etc) and focus on inferences (evolution, diversity, pedigree, etc) that have been done or can be done from the Y-chromosome. Also proofread your text for English; I am not native either, the text is readable, but there are some mistakes (I am not even commenting at those). Thanks for your efforts.

Author Response

Cardinali and colleagues overview an increasingly interesting topic, addressing the paternal evolutionary history of horses, probably the most important animal for humans.

I think the authors deserve the opportunity to fix my very major concerns. There are some statements that are not true, and some very important papers still need to be commented and referenced:

Authors’ response: we would like to sincerely thank the reviewer for the constructive comments. We have modified the manuscript by following his suggestions.

  • The adaptation to the human niche and the repeated interactions between domestic populations and surrounding wild relatives led to a continuous gene flow resulted in the current genetic variation [5,6].

In our last paper (Librado et al; Nature 2021), we demonstrated that DOM2 horses (the lineage leading to present-day domesticates) carried a diversity of mt and Y haplogroups before their expansion across Eurasia 4200 years ago. They did not interact much with wild relatives after this period. They simply replaced them. Therefore, there was no such continuous gene flow after horses left their homeland.

  • Authors’ response: we have modified the manuscript accordingly to this comment. The sentence is now the following: “The adaptation to the human niche led to the current genetic variation [5,6] carried by the domestic horses that left the Western Eurasian steppes at the beginning of the second millennium BC and moved towards Eurasia thus replacing almost all the wild populations [7]”

  • Genetic evidence pinpointed that multiple horse domestication events occurred across Eurasia 5,000 - 6,000 years ago [2,8–15]".

There was one dead-end domestication event at Botai 5,500 years ago (the so-called DOM1 horses, Outram 2009 Science; Gaunitz 2018 Science), and then another one in the lower Volga-Don region. The horse that originated from the latter expanded across Eurasia 4,200 years ago (Librado 2021; Nature). There were no multiple horse domestication events 5,000-6,000 ya. The horse lineages that were domesticated already had a diversity of haplogroups, which was originally misinterpreted as multiple domestication events. But we know now that this was not true.

  • Authors’ response: as suggested by the reviewer, we have modified the sentence as follow: “The first genetic evidence pinpointed that multiple horse domestication events occurred across Eurasia 5,000 - 6,000 years ago [2,9–16], but the first appearance of all modern horse ancestor is dated back to 4200 years ago [17]. Then, the horse domestication process was revised and the Botai horses, deeply described by Outram and colleagues [13], were considered the ancestors of modern Przewalski's horses [18], while modern horses have been domesticated in a more Western center: the lower Volga-Don region [7]”.

  • "with an increasing genetic variability that remained constant during the last 3,000 years ".

We showed that genetic diversity in the autosomes remained more or less constant in the last 4,000 years , until 250 years ago(Fages et al 2019; Cell). If we look at the Y chromosomal diversity, we see it expanded 4,200 years ago as DOM2 horses spread through Eurasia and increased their population size (Librado et al 2021). I do not understand where their sentence, and this 3000 years ago, comes from.

  • Authors’ response: we have replaced “3,000” with “4,000”.

  • "until it significantly dropped in the last ≈250 years ". This statement requires to be backed up by an original reference (eg. Fages et al). Only citing other reviews is not fair to the original publications.

  • Authors’ response: we have added the original reference Fages et al 2019.

  • "completely replacing almost all other local populations 3,000 years ago ". Again, it should be 4,000 years ago.

  • Authors’ response: we have replaced “3,000” with “4,000”.

  • "the genetic diversity among horses is very great ". What does it mean to have a great diversity? I guess the authors mean "large"? Please clarify

  • Authors’ response: we have replaced “great” with “large”.

  • "Recently , many studies have been focused on the loss of biodiversity [21–23] or the increase of deleterious genotypes [24] caused by the inbreeding, whose coefficient was calculated to evaluate the genetic diversity among different horse breeds [25–75]."

I think there is one fundamental paper missing here: Orlando and Librado 2019 published in Genes (it analyses the relation between inbreeding and deleterious mutations, as reviewed in Librado and Orlando 2021 (Annual Reviews))

  • Authors’ response: we have added the missing references. The current phrasing is: “Recently, many studies have been focused on the loss of biodiversity [24–26] or the increase of deleterious genotypes [27] caused by the inbreeding (as reviewed in [7], that in the last 200 years led to the phenotypical expression at homozygous sites of deleterious variants [28], and whose coefficient was calculated to evaluate the genetic diversity among different horse breeds [29–79]”

  • "It was finally released in 2018 , with results distributed over 31 autosomes, the X chromosome and the mitochondrial genome"

The horse genome was first released in 2009 (Wade et al.; EquCab2). In 2018, we had an improved version (EquCab3). Please clarify your text. It is confusing. It seems that the first horse assembly was produced in 2018.

  • Authors’ response: we have modified the paragraph as requested by Reviewer 3, and we have replaced “released” with “improved and updated”.

  • Their phrasing of this whole section is simply too similar to Kalbfleisch et. al 2018 (the paper presenting EquCab3), which is not acceptable

  • Authors’ response: we have modified the text accordingly to this comment and accordingly to revewer 3 also. The whole paragraph was edited and shortened.  The reference Kalbfleisch et. al 2018 is still reported

  • "As EquCab2.0 contained many gaps, the genome of Twilight was recently re-sequenced, through Next Generation Sequencing (NGS) techniques ".

EquCab3 not only required NGS techniques, but also newer sequencing technologies such as PacBio, proximity ligation libraries, etc.

  • Authors’ response: we have replaced “Next Generation Sequencing” with “high-throughput sequencing technologies”

  • "Contextually with the release of EquCab3.0 , Janečka and colleagues published the assembly of 9.5 Mb based on the sequencing of the Y chromosome from the Thoroughbred stallion “Bravo”, a half‐brother of Twilight, thus completing the Thoroughbred‐based horse reference genome and providing the first comprehensive assembly of the male‐specific region of Y chromosome (MSY) (accession number: MH341179)"

I think Lippold et al (2011; Nature Communications), Wallner et al. (2013; PLoS One), Wallner et al (2017; Current Biology) deserve to be cited in this context as well. They are cited later in the context of Y-chr diversity, but they also generated the earliest scaffolds of the Y chromosome.

  • Authors’ response: we have modified the text accordingly to this comment.

  • I would considerably reduce the section comparing the Y chromosome of horse to that of other mammals . Many paragraphs are simply descriptive, and provide nothing new conceptually. This is true in other sections, but it makes the reading particularly boring here. We have databases for consulting how many genes there are, and other features of the Y chromosome. Less is more sometimes.

  • Authors’ response: we have reduced this paragraph accordingly to the reviewer’s suggestion.

  • "In mammals, the Y chromosome is characterized by the male-specific and nonrecombining portion (MSY), inherited only along paternal lines from a father to his colts, thus becoming a large-used marker in equine studies focused on the origin of stallions [136]. "

I do not think we need this sentence again at this stage of the review (last section). We got what the Y chromosome is already.

  • Authors’ response: we have deleted this sentence.

  • "Later, Lippold and colleagues sequenced 4 kb 291 of Y chromosome of DNA derived from nine ancient (one 2,800-year-old domesticated and eight wild horses) remains retrieved in permafrost sites of Siberia and North America [142]."

You need to talk about Librado et al. 2017 (Science) here

  • Authors’ response: we have added also Librado et al. 2017 in this section.

I stopped reviewing in detail their work here. There are simply too many errors to keep going as a reviewer. I am keen on recommending this for publication, but please be correct, and unambiguous your own phrasing. Reduce only-descriptive sections (eg. how big is the assembly, how many genes, etc) and focus on inferences (evolution, diversity, pedigree, etc) that have been done or can be done from the Y-chromosome. Also proofread your text for English; I am not native either, the text is readable, but there are some mistakes (I am not even commenting at those). Thanks for your efforts.

Authors’ response: we would like to sincerely thank the reviewer for these constructive comments. We have modified the manuscript accordingly to his suggestions.

Reviewer 3 Report

The manuscript under the title: “ Unlocking the Horse Y Chromosome diversity” summarises the recent findings in the field of the molecular biology of the equine y chromosome. In general, searching for markers which could be used as valuable tools for phylogenetics, breeding history and reproduction fits in with global trends.  The presented manuscript might have publishing potential in Genes, however, requires explanations, deep editorial editing and some clarification. This is why at this stage I would recommend: reconsidering after major revision.  Overall, the manuscript is hard to read mainly to the very long sentences. I suggest deep editing of sequences for better flow.

The part in the introduction about the Horse Genome is far to long. I recommend shortening. 

Please find some detailed comments below hovewer, not all editorial corrections are included as the manuscript needs deep editing. 

ln 39: type? horses of  one origin?

Ln 46-47: this is far to simlified

Ln 48-59: should be like that, but in theory. I suggest at least delete ln 58.

2. The horse genome

Ln 71-115: the whole information at present form must be rewritten. Please see the ncbi database, genome assembly and if this part stays in manuscript should be shortened, and really updated. Manuscript is rather about Y chromosome. And lines 309-424 are really informative. 

ln 76: no, first release of horse genome was earlier. Please correct.

ln 125: sentence is unclear, please rewritte

ln 140-143: sentence is unclear, please rewritte

ln 152-154: sentence is unclear, please rewritte

ln 188-194: I am sorry, I really not sure if I understand.

ln 195-199: please claryfiy results of Jancka and please corroboratethem with th elines 204-2011becouse the information dont mach. 

The Table 2 is not really informative, and at least should be moved to suplementary material.

Author Response

The manuscript under the title: “ Unlocking the Horse Y Chromosome diversity” summarises the recent findings in the field of the molecular biology of the equine y chromosome. In general, searching for markers which could be used as valuable tools for phylogenetics, breeding history and reproduction fits in with global trends.  The presented manuscript might have publishing potential in Genes, however, requires explanations, deep editorial editing and some clarification. This is why at this stage I would recommend: reconsidering after major revision.  Overall, the manuscript is hard to read mainly to the very long sentences. I suggest deep editing of sequences for better flow.

The part in the introduction about the Horse Genome is far to long. I recommend shortening. 

Please find some detailed comments below hovewer, not all editorial corrections are included as the manuscript needs deep editing. 

Authors’ response: we would like to thank the reviewer. We have reviewed the manuscript by shortening the Horse Genome paragraph and made the reading more fluid by re-phrasing and editing the original manuscript.

  • ln 39: type? horses of  one origin?

Authors’ response: we have replaced “one type” with “those modern type horse lineages”.

  • Ln 46-47: this is far to simlified

Authors’ response: we have deleted the sentence as it was principally referred to the genetic variability caused by mare selection practice.

  • Ln 48-59: should be like that, but in theory. I suggest at least delete ln 58.

Authors’ response: we have deleted ln 58 as suggested.

  • The horse genome

Ln 71-115: the whole information at present form must be rewritten. Please see the ncbi database, genome assembly and if this part stays in manuscript should be shortened, and really updated. Manuscript is rather about Y chromosome. And lines 309-424 are really informative. 

Authors’ response: as requested the “Horse genome” paragraph was rewritten and shortened; all not necessary information was excluded.

As for lines 309-424, we did not prefer to delete the Y chromosome variation description due to the focus of the review; we have only checked English language editing.

  • ln 76: no, first release of horse genome was earlier. Please correct.

Authors’ response: the reviewer is right; we have modified accordingly.

  • ln 125: sentence is unclear, please rewritte

Authors’ response: we have simplified the sentence as follows: “The human Y chromosome was sequenced in 2003 [96], followed by chimpanzee, mouse and rhesus macaque Y chromosomes [97–99]”.

  • ln 140-143: sentence is unclear, please rewritte

Authors’ response: we have edited as follows “In horse, despite the stallion fertility has prime importance in breeding management, very little is known about the complexity of Y chromosome structure and its genetic degeneration”.

  • ln 152-154: sentence is unclear, please rewritte

Authors’ response: we have deleted the sentence.

  • ln 188-194: I am sorry, I really not sure if I understand.

Authors’ response: we have modified the sentence and omitted unnecessary details. The sentence is now: “In 2018, Janečka and colleagues generated the first comprehensive assembly and functional annotation of the euchromatic male-specific region of the Y chromosome (eMSY). They demonstrated its dynamic nature and provided a reference sequence useful to improve our knowledge about the equine male development and fertility [94].”

  • ln 195-199: please claryfiy results of Jancka and please corroboratethem with th elines 204-2011becouse the information dont mach. 

Authors’ response: we have now modified the paragraph and considerably reduced the descriptive section as also suggested by the reviewer 2.

  • The Table 2 is not really informative, and at least should be moved to suplementary material.

Authors’ response: as suggested by the reviewer we have deleted table 2. All data and details were already reported in Tables S2.

Round 2

Reviewer 3 Report

Dear Authors, 

Thank you for replying to my comments and applying suggestions. Currently, the manuscript might be considered to be published in Genes.

Some finishing touches below:

From line 311 check the reference formatting.

line 333: this sentence needs to be more informative, maybe the authors would add why?

Author Response

Dear Authors,

Thank you for replying to my comments and applying suggestions. Currently, the manuscript might be considered to be published in Genes.

Authors’ response: we would like to sincerely thank the reviewer for his comments. We have modified the manuscript by following his suggestions.

Some finishing touches below:

  • From line 311 check the reference formatting.
    • Authors’ response: we have modified the reference formatting

  • Line 333: this sentence needs to be more informative, maybe the authors would add why?

Authors’ response: we have modified the sentence as follows “In some cases, the MSY from modern horse breeds was compared with the results from Przewalski's Horse [52,91,111,133,136], which is considered the feral descendant of the domesticated Botai horse [17] and has been thoroughly investigated; as a result, a certain number of variable sites were individuated, and a horse Y-chromosomal phylogeny was generated [84,91,111].”